# Exploring Hydrogen Incorporation into the Nb_4_AlC_3_ MAX Phases: Ab Initio Calculations

**DOI:** 10.3390/ma15217576

**Published:** 2022-10-28

**Authors:** Yudong Fu, Zifeng Li, Weihong Gao, Danni Zhao, Zhihao Huang, Bin Sun, Mufu Yan, Guotan Liu, Zihang Liu

**Affiliations:** 1College of Materials Science and Chemical Engineering, Harbin Engineering University, Harbin 150001, China; 2Department of Materials Science and Engineering, City University of Hong Kong, Kowloon Tong, Hong Kong SAR, China; 3School of Materials Science and Engineering, Harbin Institute of Technology, Harbin 150001, China; 4State Key Laboratory of Advanced Welding and Joining, Harbin Institute of Technology, Harbin 150001, China

**Keywords:** Nb_4_AlC_3_ MAX phase, ab initio calculations, hydrogen storage, diffusion mechanism

## Abstract

The Nb_4_AlC_3_ MAX phase can be regarded as a TMC structure with stacking faults, which has great potential as a novel solid hydrogen storage material. Herein, we used ab initio calculations for understanding the hydrogen incorporation into Nb_4_AlC_3_ MAX phases, including equilibrium structural characteristics, energy changes, electronic structures, bonding characteristics, and diffusion paths. According to the calculated results, H has thermal stability in the interstice of the Nb-Al layer, and the most probable insertion site is an octahedron (3-site) composed of three Nb atoms and three Al atoms. When C vacancies are introduced, the Nb-C layer has a specific storage capacity for H. In addition, Al vacancies can also be used as possible sites for H incorporation. Moreover, the introduction of vacancies significantly increase the hydrogen storage capacity of the MAX phase. According to the electronic structure and bonding characteristics, the excellent hydrogen storage ability of the Nb_4_AlC_3_ structure may be due to the formation of ionic bonds between H and Nb/Al. It is worth noting that the H-Al bond in the 1-site is a covalent bond and an ionic bond key mixture. The linear synchronous transit optimization study shows that only H diffusion in Al vacancies is not feasible. In conclusion, the Nb-Al layer in Nb_4_AlC_3_ can provide favorable conditions for the continuous insertion and subsequent extraction of H, while the vacancy structure is more suitable for H storage. Our work provides solid theoretical results for understanding the hydrogen incorporation into Nb_4_AlC_3_ MAX phases that can be helpful for the design of advanced hydrogen storage materials.

## 1. Introduction

One of the key scientific and technological challenges of the 21st century is the need to meet the global demand for energy in a clean, safe, and sustainable way to overcome the problems of electricity storage and pollution [1,2,3,4]. Hydrogen is one of the most important new energy sources, and safe and efficient hydrogen storage has become an important research topic [5,6,7]. From a practical application standpoint, storing hydrogen in tanks is the most attractive method. However, the size of the tanks and losses from hydrogen liquefaction and evaporation lead to high costs [8,9]. Therefore, hydrogen storage in solid materials can serve as a compromise approach while satisfying practical application and cost requirements. It can release a large amount of chemically bound and physically adsorbed hydrogen while keeping its structure intact. The role of different solid-state materials in the adsorption process has been studied [10]. Until now, several materials have been studied, including graphene [11], metal oxides [12], zeolites [13], etc.

Transition metals can form carbides of various structures and stoichiometric ratios, depending on their position in the periodic table and the degree of defects in the structure [14]. Usually, TMCs are used as catalysts for hydrogenation, CO_2_ reduction, hydrotreating, etc. [15,16]. Tian et al. [17] found that adding NbC to MgH_2_ can significantly reduce the activation energy of the hydrogen evolution reaction. However, in recent work, transition metal carbides (TMCs) were found to have some potential to be applied for H storage and diffusion. Some researchers found that carbon vacancies and stacking faults (SFs) in TiC_x_ are very helpful for the storage and diffusion of H, and the diffusion of H in SFs layers in TiC_x_ layers is more accessible than that in regular stacked layers [18,19]. SFs in TiC_x_ can be considered as a fragile twin interface (1 1 1), and it can be found that the introduction of the SFs layer in TiC_x_ can improve the hydrogen storage capacity of TMC [20,21]. In addition to being used as a catalyst, NbC has also been shown to have a particular ability to absorb H. Salehinia et al. used molecular dynamics to obtain that H has a higher diffusion coefficient in the NbC bulk with crystal defects [22], H can exist stably in the NbC (1 1 1) plane [23].

Therefore, we speculate that NbC, TMC with certain defects, can be used as a solid hydrogen storage material. The structure of Nb_4_AlC_3_ can be regarded as a two-dimensional close-packed Al layer periodically inserted into the NbC(1 1 1) twin interface [24], which is similar to the results of Ti_3_AlC_2_ [25], and both of them belong to MAX phase materials. The chemical formula of such ternary layered materials with similar structures and properties can be represented by M_n+1_AX_n_ (*n* = 1, 2, 3), where M is an early transition metal, A is a group A element, and X is carbon or nitrogen [26]. The MAX phase exhibits beneficial properties of metals, such as machinability and excellent thermal and electrical conductivity. It exhibits good properties of ceramics, such as high modulus, low density, and oxidation resistance [27,28,29]. In addition, according to this speculation, Nb_4_AlC_3_ is used in the field of hydrogen storage.

In our previous study, the Nb_4_AC_3_ MAX phase material was comprehensively investigated using ab initio calculations, including electronic structure, optical properties, and anti-oxidation properties. The calculation results show that the Nb_4_AC_3_ MAX phase has a higher bulk modulus and lower compressibility [30]. Moreover, Nb_4_GaC_3_ and Nb_4_SiO_3_ have good anti-oxidation properties, which are attributed to the special bond and relationship between Si and O (mixed bond of the covalent bond and ionic bond) [31]. Some studies have used ab initio calculations to explore the mechanism of H incorporation into MAX phase materials [32]. To verify our hypothesis, this paper also uses density functional theory (DFT) to explore the hydrogen storage principle of the MAX phase at the atomic scale. This work investigates the doping of H in the Nb-Al layer and the Nb-C layer, followed by the introduction of Nb, C, and Al vacancies. In addition, the diffusion mechanism of H in the Nb-Al layer, Nb-C layer, and vacancies was also studied.

## 2. Computation Details

All calculations were performed using the CASTEP (Cambridge Sequential Total Energy Package) code [33] based on density functional theory (DFT), Vanderbilt-type ultrasoft pseudopotential (USPP) was chosen by us to describe the frozen core approximation [34], the cross-correlation by Perdew–Burke–Ernzerhof (PBE) parametric generalized gradient approximation (GGA) [35,36,37]. To ensure calculation accuracy, the cutoff energy of the plane wave was set to 450 eV, and the Monkhorst–Pack scheme of the particular k point was selected for Brillouin zone sampling [38]. The cut-off energy and k-mesh grid were tested strictly. The convergence criteria for structure optimization were energy change of all atoms: <5 × 10^−6^ eV/atom, maximum force < 0.01 eV/Å, maximum displacement < 5 × 10^−4^ Å, maximum stress < 0.02 GPa.

A linear synchronous transport (LST) method was employed in the study of the H atom diffusion. In LST optimization, single-point energy calculations were performed on a set of linearly interpolated structures between the initial and final states. The maximum energy structure along this path provides the first possible transition state, followed by energy minimization in the direction of conjugation of the reaction path [25,39]. This method can be used to determine the energy barrier for H diffusion.

## 3. Results and Discussion

### 3.1. Incorporation Properties of the Hydrogen Atom in Nb_4_AlC_3_

Based on previous studies [31], the hydrogen storage capacity of the system is influenced by the following factors. First, the different atomic configurations around the doped H atom (simplified as H) manifest as H atoms occupying different positions as foreign interstitial atoms (FIA). The second is the energy change before and after the incorporation of H. Then, there is the interaction of H and surrounding atoms. Finally, according to the characteristics of hydrogen storage materials, the diffusion mechanism of H in the structure also needs to be studied. Therefore, we will study the hydrogen storage capacity of the Nb_4_AlC_3_ MAX phase from the above aspects, which is like our previous study on the oxidation capacity of the MAX phase [31]. The space group number of Nb_4_AlC_3_ is P63/mmc, and it has a layered hexagonal structure whose configuration is like the periodic insertion of monolayer Al atoms in the NbC (1 1 1) plane.

Therefore, the possible sites in this structure mainly exist in the Nb-C and Nb-Al layers, as shown in Figure 1. There are three possible sites for H in the Nb-Al layer. The first is the 1-site (tetrahedron), composed of three Nb atoms and one Al atom. Then there is the 2-site (tetrahedron), which consists of three Al atoms and one Nb atom. Finally, the 3-site (octahedron) consists of three Al atoms and three Nb atoms. Only tetrahedra exist in the Nb-C layer because C atoms occupy the octahedral centers composed of Nb atoms. According to crystal symmetry, there are three sites for H existing in the Nb-C layer, the first is the 4-site (tetrahedron) composed of one Nb(1) atom and three Nb(2) atoms, and the second is the 5-site (tetrahedron) composed of three Nb(1) atoms and one Nb(2) atom. The last is the 6-site (tetrahedron) composed of four Nb(2) atoms. It is worth noting that the distance between the Nb(1) and Nb(2) atomic layers is shorter than that between the Nb(2) and Nb(2) atomic layers, which indicates that the 6-site has more space. After the convergence test, it is found that the 2 × 2 × 1 supercell can reach convergence with reasonable accurateness, which is in agreement with previous works [32,40]. Taking H incorporation into 3-site as an example, the calculated *E_for_* is −0.53 eV for 2 × 2 × 1 supercell, −0.54 eV for 2 × 3 × 1 supercell, and −0.56 eV for 3 × 3 × 1 supercell, respectively. As a larger size supercell will cause much more computational time consumption, the 2 × 2 × 1 Nb_4_AlC_3_ supercell is used in this work.

Then, the hydrogen storage capacity can be measured by *E_for_*. The formula for *E_for_* is as follows [32]:(1)$$E_{for}=E_{tot}^{H}-E_{tot}-n\frac{E^{H_{2}}}{2}$$
where $E_{tot}^{H}$ is the system’s total energy with *H*, $E_{tot}$ is the total energy of Nb_4_AlC_3_, and $E^{H_{2}}$ is the total energy of the hydrogen molecules. Then, vacancies are introduced in Nb_4_AlC_3,_ and the formula for $E_{form}^{H-v}$ is as follows:(2)$$E_{form}^{H-v}=E_{tot}^{H-v}-E_{tot}^{v}-n\frac{E^{H_{2}}}{2}$$
where $E_{for}$ is the total energy of the Nb_4_AlC_3_ with a vacancy and an H atom, $E_{tot}^{v}$ is the total energy of Nb_4_AlC_3_ with a vacancy. In addition, the structural stability of the H-incorporated system is confirmed by the cohesive energy $E_{coh}$ [41]:(3)$$E_{coh}=E_{total}-E_{atoms}$$
where $E_{total}$ and $E_{atoms}$ are the calculated total energy of the system and the total energy of all the constituent atoms in the isolated state, respectively.

The calculation results are shown in Table 1 and Figure 2, including the formation energy, cohesive energy, and lattice constant of the different systems: Nb-Al layer (1-, 2-, 3-sites) and Nb-C (4-, 5-, 6-sites) layer.

Among them, a lower negative *E_coh_* means better formation ability, while a positive *E_coh_* means that the system cannot be stable at absolute 0 K. And the relaxed structure is shown in Figure 3. It can be seen from Figure 2 that the cohesive energies of all structures increased after the incorporation of H but were less than 0, which reflected the system’s thermodynamic stability after the incorporation of H was reduced. However, it still existed stably at absolute 0 K. Figure 3a,b show the 1-site relaxed structure. It can be found that H is in a tetrahedron interstice, only the c-axis of the structure increases slightly, and the corresponding *E_for_* is −0.50 eV. This suggests that the 1-site is able to incorporate H. Figure 3c,d show the 2-site’s relaxed structures. H no longer exists in the tetrahedron composed of one Nb atom and three Al atoms, but it is close to the three Al atoms. At this time, H is located near the center of the hexahedron, composed of two Nb(1) atoms and three Al atoms. Moreover, the changes in lattice parameters are a b-axis elongation and c-axis shortening, which are consistent with the structural features after relaxation. The *E_for_* of this structure is −0.46 eV, which is higher than that of the 1-site. This is attributed to the weaker H stability in the large-sized the interstitial space [42]. The 3-site relaxed structure is shown in Figure 3e,f, where H is in the octahedral center composed of three Al atoms and three Nb atoms. The a-axis is more elongated than the b-axis, and the c-axis is unchanged. The *E_for_* of this structure is −0.53 eV, which is slightly lower than that of the 1- and 2-sites. Therefore, the 3-site is the best incorporation site of H in the Nb-Al layer.

Next, the sites for possible incorporation of H in the Nb-C layer were analyzed. The 4-site relaxed structure is shown in Figure 3g,h, where H is in the center of the tetrahedron composed of four Nb atoms. This structure is a, and b axes have the same degree of growth. The c-axis is shortened. It is worth noting that the *E_for_* is positive, so the stability of H in the 4-site is poor. The 5-site relaxed structure is shown in Figure 3i,j. It can be found that the 5-site is not a stable hydrogen storage site because the H atom enters 1-site after structural relaxation, which further proves that the Nb-Al layer has a good absorption capacity for H.

Finally, the 6-site relaxed structure is shown in Figure 2k,l. It can be found that the H atom moves to the Nb(2) atomic layer after relaxation and is located between three Nb(2) atoms. This phenomenon may be because the 6-site is a larger tetrahedral interstice than the 4-site. Therefore, we speculate that in the same kind of interstitial with enough interstitial space to accommodate H, the stability of H atoms in smaller interstitials would be higher. In [42], H is the most stable in the tetrahedral interstice of close-packed hexagonal pure titanium, and the tetrahedron is the smallest interstice in this structure. Therefore, the H atoms move into the Nb(2) layer instead of being stable in the tetrahedron (6-site). It is worth noting that the distance between H and a C(2) atom is very close, and it can be speculated that H and the C (2) atom are bonded. However, the value of the *E_for_* of 6-site is positive. Hence, 6-site is also not stable for H to exist.

According to the above results, the thermodynamic stability of H in the Nb-Al layer is higher than that in the Nb-C layer, and the *E_for_* of the 3-site is the lowest. Hence, the 3-site has the most robust incorporation ability of H. This phenomenon is because 3-site (octahedron) has more space than 1-site and 2-site (both tetrahedrons).

The H storage capacity of TMC will increase after the introduction of crystal defects. Therefore, the H doping of the Nb_4_AlC_3_ MAX phase structure with vacancies is studied and analyzed next. According to the crystal symmetry, the Nb atoms and C atoms in the Nb-C layer have two kinds of occupied sites, respectively, as shown in Figure 1a. So, there are five kinds of vacancies for H incorporation, namely Nb1-v, Nb2-v, Al-v, C1-v, and C2-v, as shown in Figure 4.

The hydrogen storage capacity of the structure is still reflected in the *E_for_* and *E_coh_*, as shown in Figure 6. The structures incorporating H still all have negative *E_coh_* values, indicating that these structures are thermodynamically stable at absolute 0 K.

The structures after the relaxation of several different vacancies absorbing H are shown in Figure 5. In the calculation process, we find that H cannot exist stably in the Nb(1) vacancy and move to the site composed of one Al atom and three Nb(1) atoms (the NbC layer below) in the tetrahedron, which means that Nb(1) vacancies cannot be used as storage sites for H. It can be seen from Figure 5g,h that H is also not stable in the Nb(2) vacancy and will combine with the adjacent C(1) atom, which is similar to the case of H in the 6-site, as shown in Table 2. Therefore, H is less likely to be present in some lattice vacancies, probably because these vacancies are much larger than H atoms, so H is somewhat more inclined to be in interstitial sites. Then, as shown in Figure 5e,f, H can stably exist in the Al vacancy with an *E_for_* of −2.41 eV, and the c-axis of the structure can be found to be shortened because the atomic radius of H is much smaller than that of Al. Finally, as shown in Figure 5a–d, H can exist stably in both C(1) vacancies and C(2) vacancies; the *E_for_* are −3.05 eV and −3.68 eV, respectively. And the lattice constants of the a, b, and c axes are shortened. It is worth noting that the relaxed position of H in C(1)v is located in the center of the octahedron consisting of six Nb(2) atoms.

In contrast, the relaxed position of H in C(2)v is located below the center of the octahedron. This is because C(2)v is located in a less symmetrical octahedron composed of three Nb(1) atoms and three Nb(2) atoms, and the distances from the two vertices to the center of the octahedron are 2.177 Å and 2.233 Å, respectively. The *E_for_* of the Ti_3_AlC_2_ structure with a C vacancy is −1.16 eV, as shown in Figure 6. This indicates that the Nb_4_AlC_3_ structure with a C vacancy has a higher hydrogen absorption capacity.

Many studies have demonstrated that C vacancies in transition cermets have a solid ability to trap H [25,43]. However, there are relatively few studies on trapping H by Al vacancies. Thus, we calculated the *E_coh_* of the five vacancy models using Formula (2). Table 3 shows the *E_coh_* of different structures with vacancies, among which the *E_coh_* of the Al vacancy structure is the lowest, even lower than the initial structure. It can be speculated that there is a high probability of Al vacancies in the Nb_4_AlC_3_ MAX phase structure. While the cohesive energies of the Nb(1)v structure and Nb(2)v structure are relatively high. Therefore, in the Nb_4_AlC_3_ MAX structure, Al, C(1), and C(2) vacancies are more likely to appear. Based on our calculation results, we believe that C(1), C(2), and Al vacancies positively affect the incorporation of H. Hence, A, X element (MAX) vacancies are more likely to appear in the MAX phase structure, and the same situation has also occurred in previous studies [43].

### 3.2. Bonding Behavior of the H-Doped Nb_4_AlC_3_

The structure and properties of materials are mainly reflected in the electronic structure. Therefore, we describe the bonding properties of the structure by the density of states, the charge density difference, and the Mulliken population. Figure 7 shows the DOS (Density of states) and PDOS (Partial density of states) of the Nb_4_AlC_3_ structure incorporating H, and Figure 7b–d shows the DOS and PDOS of 1, 2, and 3-sites, respectively, and Figure 7e,f show the DOS and PDOS of the 4-site and 6-site, respectively. Moreover, it was found that H cannot exist stably in Nb(1)v and Nb(2)v structures, but it can exist stably in Alv, C(1)v, and C(2) v structures. Therefore, we calculated and studied the electronic structures of Al v, C(1)v, and C(2)v structures, and the calculation results are shown in Figure 7g–i. Before systematically studying the density of states of the doped H structure, we calculated the DOS for the 2 × 2 × 1 supercell of Nb_4_AlC_3_, as shown in Figure 7a.

The DOS of each doped structure shows a positive value at the Fermi level, and the Nb-4d state is still dominant. Therefore, the Nb_4_AlC_3_ MAX phase still maintains the original crystal structure after H doping. Firstly, the DOS of the H doping system in the Nb-Al layer is analyzed. The DOS of the 1-site is shown in Figure 7b, in the region far from the Fermi level, the peak at −11.3 eV in DOS is contributed by the s state of C and the d state of Nb, which indicates the formation of Nb-C from the hybridization between Nb and C. Then, the region from −5 eV to −4 eV is contributed by the d-state of Nb and the p-state of C, which is the typical TM_4d_-C_2p_ bonding relationship of the MAX phase. This is consistent with our previous research [30]. The peak at −9.1 eV in DOS is contributed by the s-state of H, the s-state of Al, and the 4d state of Nb, indicating that the formation of Al-H and Nb-H bonds comes from the hybridization of H with Al and Nb, respectively. The density of states of H incorporated into the 2-site, and 3-site are shown in Figure 7c,d, respectively. It can be found that it is almost the same as that at the 1-site, except for the decrease of the PDOS peak of H, which indicates that the hybridization of H and surrounding atoms is weakened. Then, the comparative analysis of the density of states of H in the Nb-C layer is carried out. The density of states at the 4-site is shown in Figure 7e. The peak at −8.5 eV in DOS is contributed by the s-state of H and the d-state of Nb, which illustrates the formation of Nb-H bonds. The density of states at the 6-site is shown in Figure 7f. The peak at −14 eV in DOS is contributed by the s-state of C and the s-state of H, which indicates that the formation of the C-H bond originates from the hybridization between C and H. It is worth noting that hybridization occurs between Nb and C in the 6-site, thus, the absorption of H at the 6-site is mainly attributed to the bonding of the original C(2) and H.

Finally, the DOS of these three doped structures with vacancies all show positive values at the Fermi level, and the Nb-4d state is still dominant, as shown in Figure 7g–i. In the region far from the Fermi level, the peak at −11.3 eV in DOS is contributed by the s-state of C and the d-state of Nb, which indicates that the formation of Nb-C arises from the hybridization between Nb and C. Then, the region from −5 eV to −4 eV is contributed by the d-state of Nb and the p-state of C, which is the typical TM d-C p-bonding relationship of the MAX phase. Then, Figure 5g shows the DOS of H incorporated into the C(1)v structure, and the PDOS of H shows a peak at 6.74 eV above the Fermi surface, contributed by the s-state of H and the d-state of Nb. This shows that when the Nb-H bond is formed in the C(1)v structure, this structure has specific stability. The DOS diagram of the H-doped structure of C(2)v is shown in Figure 5h. The −3–1 eV part below the Fermi surface of the DOS is contributed by the s-state of H and the 4d-state of Nb, which explains that the Nb-H bond formation originates from the hybridization of Nb and H, and the same situation occurs in the Al(v) structure, as shown in Figure 5i.

To further obtain the bonding characteristics of H atoms and surrounding atoms, we studied the differential charge density (CDD) along the (1 1 1) crystal plane of the H-doped Nb_4_AlC_3_ MAX phase, and the results are shown in Figure 8.

The CDDs where H is incorporated at 1-, 2-, and 3-sites in the Nb-Al layer are shown in Figure 8a–c, and the CDDs where H is incorporated at 4- and 6-site in the Nb-C layer are shown in Figure 8d,e, the vacancy structure is shown in Figure 8f–h. In the 1-, 2-, and 3-sites, there is partial electron transfer between H atoms and surrounding Nb and Al atoms, showing characteristics of ionic bonds. The amount of charge transfer at the 1-site is the highest in the Nb-Al layer. The H and other atoms in the Nb-Al layer only form bonds with ionic character, which is different from our previous case of adding O in the Nb-A layer [31] because the electronegativity of H is much less than that of O. Similar to the case of the Nb-Al layer at the 4-site, charge transfer occurs between the H and the surrounding Nb atoms, showing an ionic bond characteristic. The charge transfer in the 6-site does not appear between H and surrounding Nb atoms but concentrates between H and a neighboring C (2) atom, forming a C-H covalent bond. However, the electronegativity is close, and the electron Clouds are biased toward carbon atoms. As in the previous analysis, the 6-site is a tetrahedron with a larger space than the 4-site, and H may not be able to exist stably in a larger space. Therefore, higher stability will be obtained by combining with C. Then, in the three vacancy-containing structures, a certain amount of charge transfer occurred between H and surrounding Nb atoms, indicating that a bond with an anionic character is formed between H and Nb. It is worth noting that the electron density difference of H in the Al(v) structure hardly changes, which is different from the Al atoms present at this position in the parent phase. This phenomenon is because H atoms have higher Pauli electronegativity than Al atoms and are more likely to obtain electrons. In addition, it can be observed that the charge transfer before Nb and H is significantly less than that between Nb and Al. This shows that the bonding strength between Nb and H is low, which provides evidence for increased cohesive energy after H doping into the Alv structure.

Moreover, in all H-doped structures, the presence of a small number of electrons between Nb-Al indicates the formation of metallic bonds, indicating that the doping of H does not change the metallic character of the MAX phase. Then, the Nb and C atoms in the Nb-C layer show the characteristics of covalent bonds, and the properties of the MAX phase structure are determined mainly by the M-X bonding [30]. Hence, the doping of H does not significantly influence the structure, which again proves that the doping of H does not reduce the structural stability.

Table 4 shows the number of Mulliken overlap populations P and bond lengths for H doping systems. A lower *p*-value represents a very high degree of ionic character, with values close to and equal to 0 representing pure ionic bond formation. In contrast, relatively large positive values represent a covalent character. To reveal the characteristics of the bonds, the group ionic degree can be obtained using the concept of the “ionic degree scale” proposed by He et al. [44], which is defined as
(4)$$P_{i}=1-exp\left(-\frac{\left|P_{c}-P\right|}{P}\right)$$
where *Pc* corresponds to the *P* of a fully covalent bond (in this study, we assumed *P_c_* = 1). Using the above formula, *P_i_* = 1 and *P_i_* = 0 correspond to pure ionic bond and pure covalent bond, respectively. The *P* and *P_i_* of different chemical bonds are shown in Figure 9 and Table 4. According to the *P* of the 1-site structure, it is found that pure ionic bonds are formed between the H atom and Nb atoms since the P of Nb-H is 0. It can be seen from Figure 9 that a mixed bond of ionic bond and a covalent bond is formed between the H atom and Al atoms. When H is doped at the 2-site, the *p*-value of the bond between H and Al/Nb are all positive, which indicates that Nb/Al and H have bonds, respectively. It is worth noting that the distances from the H atoms to the two endpoints of the hexahedron are unequal, so H in the relaxed structure is not in the center of the hexahedron. According to Figure 9, both Nb-H and Al-H bonds are mixed bonds composed of ionic and covalent bonds. Then, when H exists in the 3-site, the H-Nb bonds are more covalent than the H-Al bonds, opposite to the situation when H exists in the 1- and 2-sites. This bonding feature enables H to exist stably in the larger octahedral interstice. At the same time, it can be found that the distribution of bond length between H, Nb, and Al is consistent with the octahedral structure. Then, H only binds to Nb(1) atoms at the 4-site, and the H-Nb bonds are pure ionic bonds, which results in the weak stability of H at the 4-site. The H atom in the 6-site moves out of the tetrahedron composed of four Nb(2) atoms after relaxation and exists stably near the C(2) atom. According to Table 4, the *p*-value of the H-C(2) bond is positive and much higher than that of the H-Al and H-Nb bonds in other sites. According to Figure 9, the bond formed by the H and C(2) atoms is strongly covalent, consistent with the previous analysis.

According to the previous analysis, H can exist stably in C(1)v, C(2)v, and Alv. It can be seen from Figure 9 that the H atom in C(1)v and the surrounding six Nb(2) atoms form pure ionic bonds. When H is present in C(2)v, the bond between the H atom and the Nb(2) atom is also pure ionic, and the bond with the Nb(1) atom exhibits a less covalent character. It can be seen from Table 4 that the bond lengths of H-Nb(1) and H-Nb(2) are 2.103 Å and 2.334 Å, respectively. The shorter bond length decreases the ionic bond characteristics of the H-Nb(1) bond. When H exists in Al v, H is located between six Nb(1) atoms, and the Nb-H bond length is equal. The Nb(1) atoms come from two different Nb-C layers in the Nb_4_AlC_3_ MAX phase structure. This site is the same as the B site for analyzing the O doping of O atoms in the MAX structure [31]. Finally, it can be found that pure ionic bonds are formed between H and Nb(1) atoms. In addition, the P of Nb-C bonds in all systems has not changed after H doping, confirming that H doping has little effect on the entire Nb_4_AlC_3_ MAX phase.

### 3.3. The Diffusion Mechanism of Hydrogen Atoms in Nb_4_AlC_3_ Structures with and without Vacancies

The research on the hydrogen storage capacity of the Nb_4_AlC_3_ MAX phase cannot only focus on the adsorption of H. The study of FIA in materials should consider its diffusion behavior [45]. According to the calculation results of system energy change and bonding properties, it can be known that H has three stable sites in the Nb-Al layer. However, it cannot exist stably in the Nb-C layer, only the introduction of C vacancies will change this situation. However, H does not necessarily have weak diffusivity in NbC. Thus, when studying the diffusion of H, we consider the following diffusion paths. Nb-Al layer diffusion path (shown in Figure 10a): from 1-site to adjacent 2-site (path I), then from 2-site to 3-site (path II), and finally from 3-site to another 1-site (path III). Nb-C layer diffusion path: from 4-site to another adjacent 4-site (path IV, shown in Figure 8a). Diffusion paths for structures with vacancies: Diffusion from a C1(C2, Al) vacancy to another adjacent C1(C2, Al) vacancy (paths V, VI, and VII, as shown in Figure 10b).

The calculation results of the transition state of the diffusion of the Nb-Al layer are shown in Figure 10c. It can be found that the diffusion energy barrier is the highest when H is interdiffused between the 1- and 2-sites, which is 0.74 eV and 0.69 eV, respectively. Combined with the previous bonding calculation results, a strong covalent bond is formed between H and Al atoms. Therefore, the migration of H atoms must cross a higher energy barrier, like other structures containing Al atoms [25].

When the diffusion path involves the 3-site, the diffusion energy barrier is significantly reduced, whether it is the diffusion between the 2-site and the 3-site or the diffusion between the 1-site and the 3-site. Especially when diffusing from the 2-site to the 3-site, the diffusion barrier is only 0.27 eV. This is because the almost pure ionic bond is formed between the H atom and Al/Nb atoms in the 3-site, which causes a weaker binding ability for H and increases the mobility of H. Hatano Y et al. [46] studied the diffusion of H in TiC with a measured activation energy of 0.90 ± 0.13 eV. In addition, the corresponding H diffusion activation energy after adding NbC as a catalyst into MgH_2_ is 1.02 eV [17]. It can be found that the activation energy for the diffusion of H in the Nb-Al layer is the lowest. Therefore, the easy diffusion of H in the Nb-Al layer will facilitate the continuous insertion and subsequent extraction of H in the Nb_4_AlC_3_ MAX phase.

According to the above analysis results, we speculate that the most probable diffusion mode of H in the Nb-Al layer is that, with the 3-site as the intermediate position, the H atom in the 1- or 2-site first diffuses to the 3-site, and then diffuses from the 3-site to the adjacent 1- or 2-site. This kind of diffusion mode with an octahedron as the middle position also appears in the study of hydrogen storage in the Ti_3_AlC_2_ MAX phase [25].

Then, it can be seen from Figure 10d that the diffusion energy barrier of path IV in the Nb-C layer is only 0.28 eV. The calculated transition state structure is similar to the 6-site. Combining H and the C(2) atom results in a transition state structure with low energy. Therefore, although the stability of H in the 4-site is poor, the diffusion from the 4-site to the adjacent 4-site occurs relatively easily, which also provides theoretical support that NbC can be used as a catalyst [17].

Finally, we investigate structures with vacancies, and model and calculate H diffusion in a structure pre-set with two vacancies. During the calculation of H diffusion in Al vacancies, we find that when H moves out of Al vacancies, it cannot continue to exist stably in the Al layer because the interstice between the two Nb(1) layers is significant, and there are few H stable sites. Although H can exist stably in the Alv vacancy, it cannot diffuse. Therefore, H is not suitable for storage on this site. The diffusion barriers of H in C1 vacancies and C2 vacancies are 0.77 eV and 0.62 eV, respectively, as shown in Figure 10e,f. Combined with the previous formation energy calculation results, C vacancies can also be used as stable sites for the continuous insertion and subsequent H extraction in the Nb_4_AlC_3_ MAX phase.

In summary, the transition state calculation results of H diffusion at the 4-position in Nb-C provide some insights and theoretical support for NbC as a catalyst. Then, H can exist stably in the Nb-Al layer, and C vacancies and the migration can be completed. Among them, the Nb-Al layer has a strong release ability of H, and C vacancies are more suitable for the stable existence of H due to their low formation energy. Therefore, the Nb_4_AlC_3_ MAX phase structure can be used as solid hydrogen storage material in the field of hydrogen storage.

## 4. Conclusions

Ab initio calculations investigated the incorporation and diffusion of H in the Nb_4_AlC_3_ MAX phase, and the following hydrogen incorporation sites were investigated: 1-, 2-, and 3-sites in the Nb-Al layer, and 4-, 5-, and 6-sites in the Nb-C layer and structures containing vacancies. The hydrogen storage mechanism of the MAX phase is analyzed in detail from the aspects of equilibrium structure characteristics, energy changes, electronic structure, bonding characteristics and diffusion paths, and the following conclusions were obtained:(1)The incorporation of H hardly changes the lattice constant. The Nb-Al layer can provide favorable sites for the storage of H. The Nb-C layer can only store hydrogen when C vacancies are introduced. In addition, H in the Al vacancies can also exist stably, but the Nb vacancies cannot be used as hydrogen storage sites.(2)H and the surrounding metal atoms form bonds with ionic characteristics; however, the H-Al bond in the 1-site is a mixed bond of an ionic bond and a covalent bond. In addition, the H in the 6-site tends to bind to the C(2) atom, and forming H-C(2) bond is almost purely covalent.(3)In the Ti-Al layer, the 3-site can be used as an intermediate site to achieve effective diffusion. The Ti-Al layer can provide favorable conditions for the continuous insertion and subsequent H extraction into Nb_4_AlC_3_. Although C vacancy has a substantial storage capacity for H, the diffusion capacity of H in it is relatively weak.(4)Interestingly, the energy barrier of H diffusion at the 4-site in the Nb-C layer is low. This can provide some theoretical support for NbC as a catalyst to promote hydrogen absorption and desorption.

## Figures and Tables

**Figure 1 materials-15-07576-f001:**
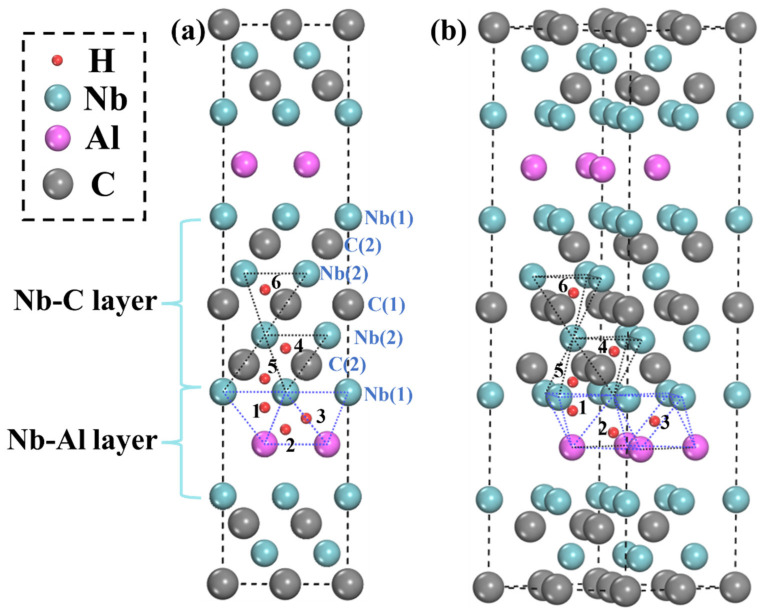
Crystal structure of Nb_4_AlC_3_ and interstitial sites considered in this work: (**a**) Front view; (**b**) side view.

**Figure 2 materials-15-07576-f002:**
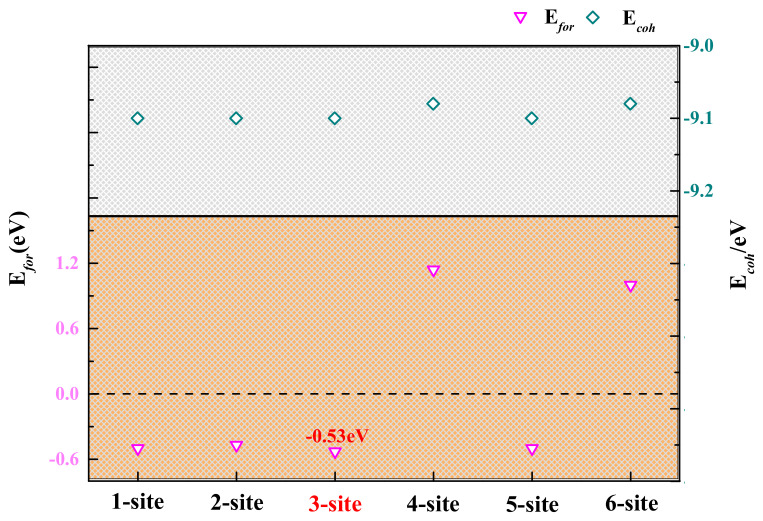
*E_coh_* and *E_for_* for H incorporation into different interstices of Nb_4_AlC_3_.

**Figure 3 materials-15-07576-f003:**
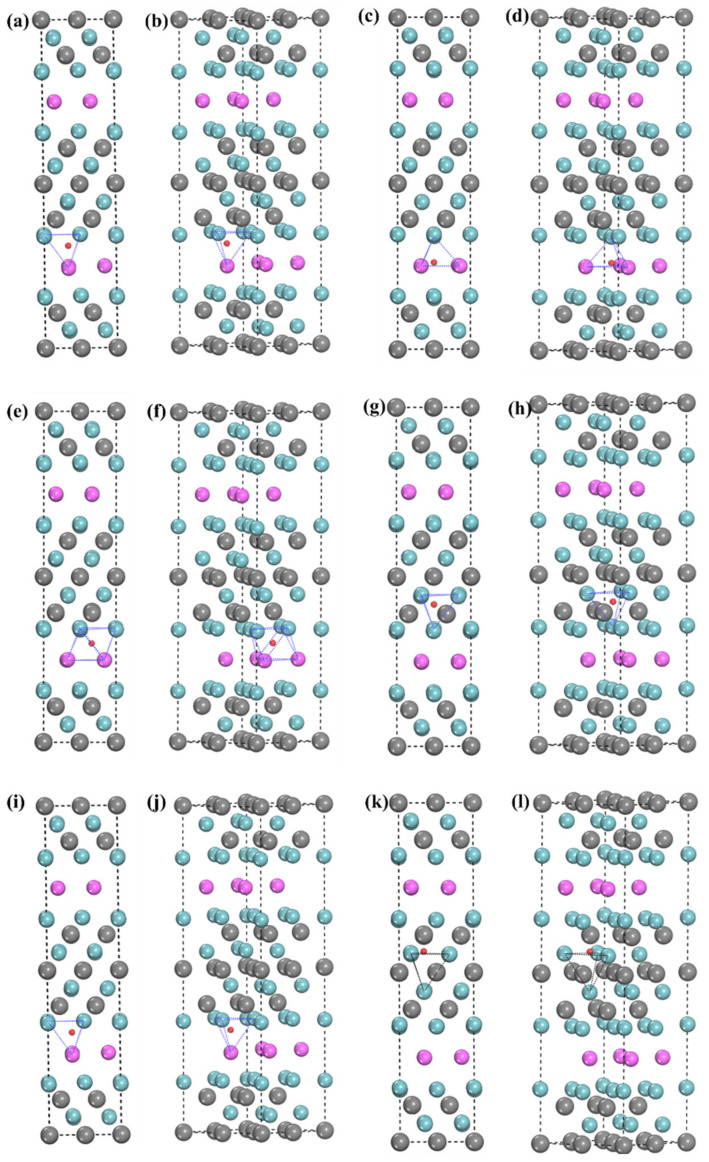
Lowest-energy structures for H incorporation into different interstices of Nb_4_AlC_3_ (**a**,**b**) 1-site; (**c**,**d**) 2-site; (**e**,**f**) 3-site;(**g**,**h**) 4-site; (**i**,**j**) 5-site; (**k**,**l**) 6-site.

**Figure 4 materials-15-07576-f004:**
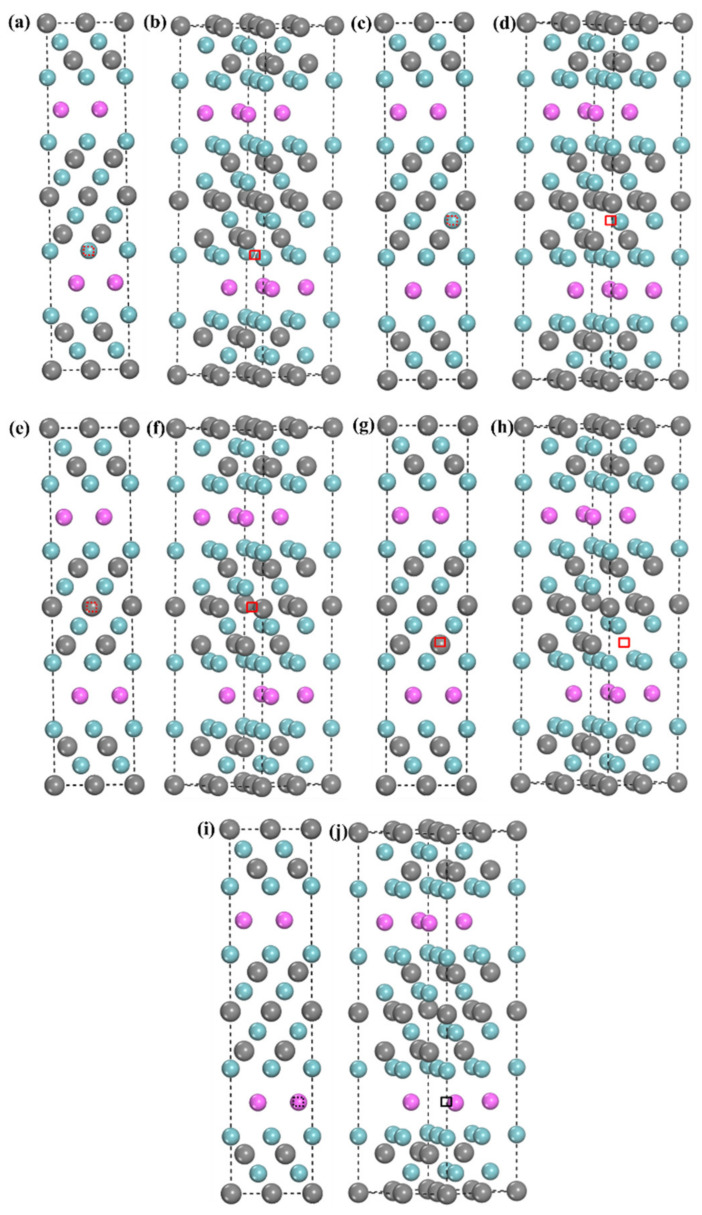
Crystal structure of Nb_4_AlC_3_ with vacancies considered in this work (**a**,**b**) Nb(1)v; (**c**,**d**) Nb(2)v; (**e**,**f**) C(1)v; (**g**,**h**) C(2)v; (**i**,**j**) Al(v).

**Figure 5 materials-15-07576-f005:**
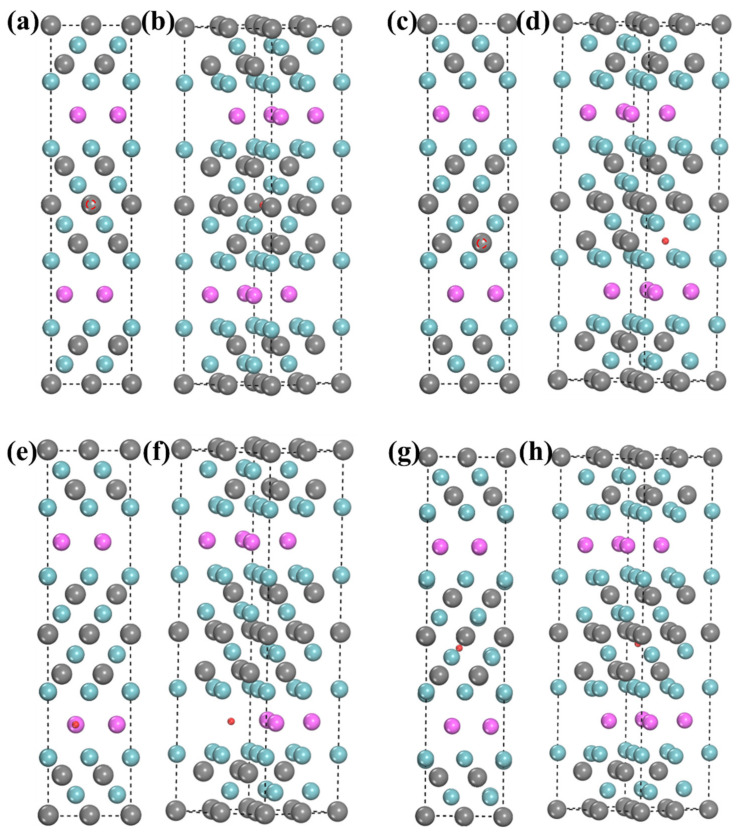
Lowest-energy structures for H incorporation into different vacancies of Nb_4_AlC_3_ (**a**,**b**) C(1)v; (**c**,**d**) C(2)v; (**e**,**f**) Alv;(**g**,**h**) Nb(2)v.

**Figure 6 materials-15-07576-f006:**
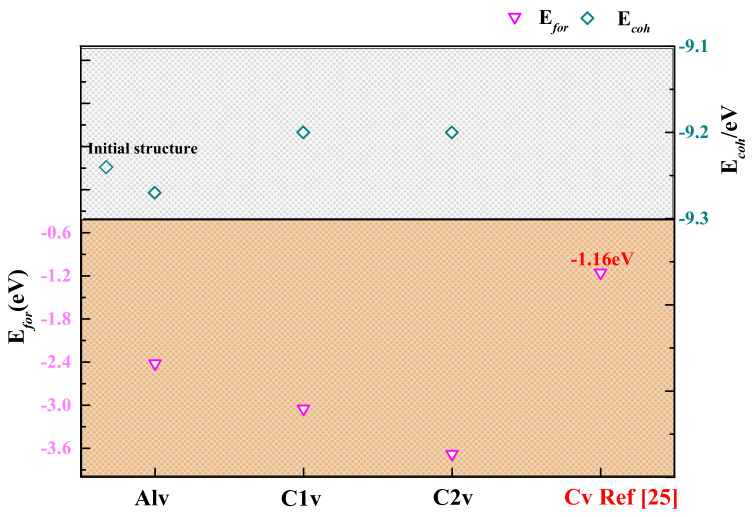
*E_coh_* and *E_for_* for H incorporation into different vacancies of Nb_4_AlC_3_.

**Figure 7 materials-15-07576-f007:**
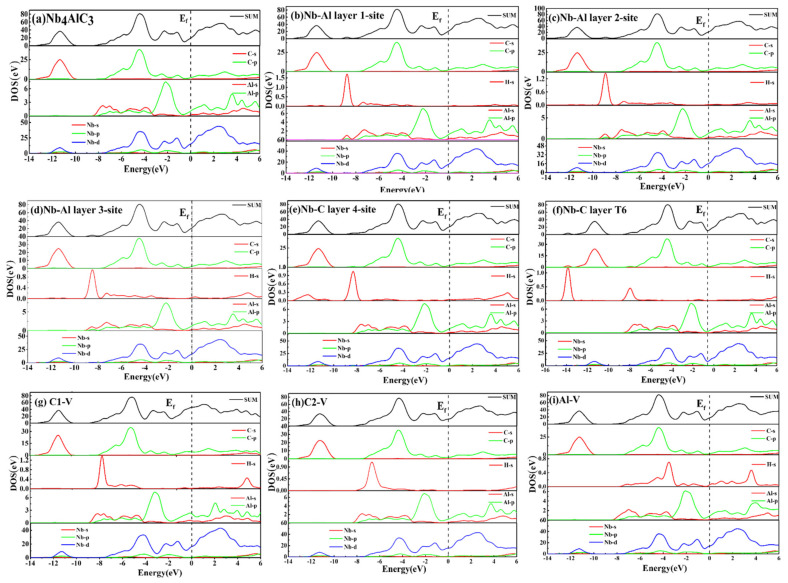
The density of states (DOS) of the H-doped Nb_4_AlC_3_.

**Figure 8 materials-15-07576-f008:**
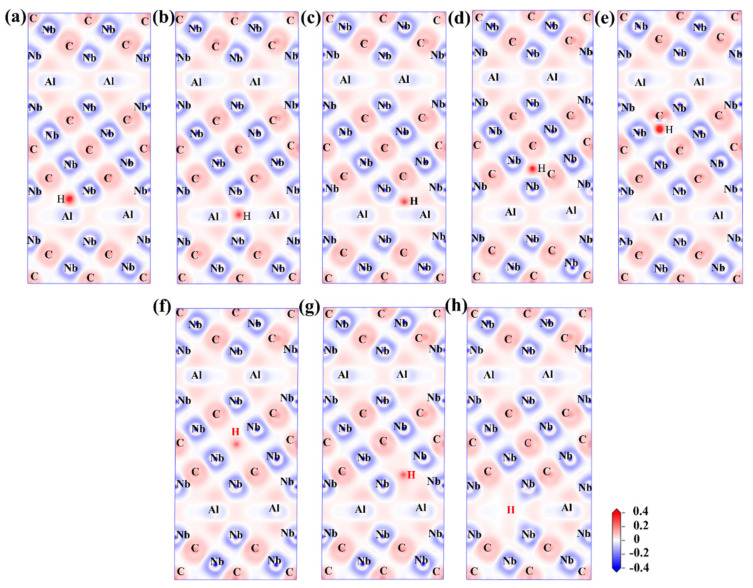
Two-dimensional charge density difference (CDD) plots in the (1 1 1) plane of H-doped Nb_4_AlC_3_ MAX phases: (**a**) 1-site; (**b**) 2-site; (**c**) 3-site; (**d**) 4-site; (**e**) 6-site; (**f**) C(1)v; (**g**) C(2)v; (**h**) Alv.

**Figure 9 materials-15-07576-f009:**
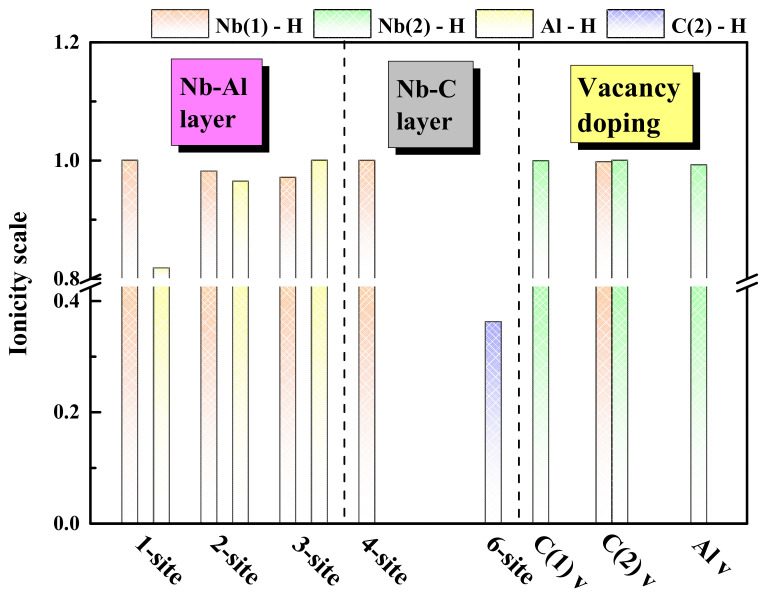
Ionicity scale Pi of Nb, Al, C–H bonds in Nb_4_AlC_3_ MAX phases.

**Figure 10 materials-15-07576-f010:**
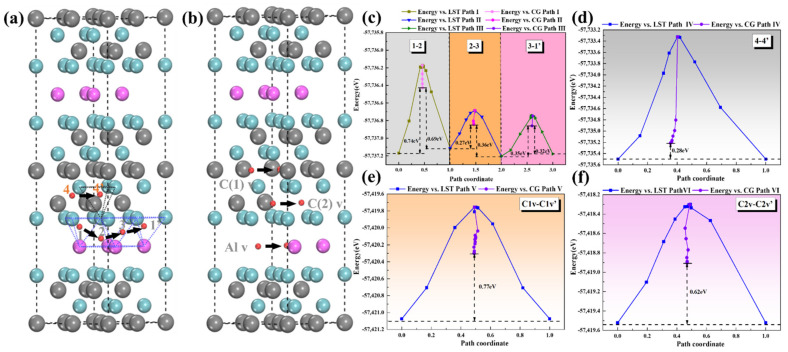
Diffusion of H in Nb_4_AlC_3_ (**a**,**b**); energy profiles and diffusion energy barrier (eV) (**c**–**f**).

**Table 1 materials-15-07576-t001:** The changes of crystal structure for H incorporation into different interstices of Nb_4_AlC_3_.

Site		Length a (Å)	Length b (Å)	Length c (Å)	Volume (Å^3^)
Initial	Final				
Initial structure		3.146	3.146	24.34	208.62
1	1	3.149	3.149	24.38	209.28
2	2	3.151	3.151	24.27	208.74
3	3	3.157	3.148	24.34	208.83
4	4	3.153	3.153	24.32	209.36
5	1	3.149	3.149	24.38	209.28
6	6	3.156	3.156	24.29	209.53

**Table 2 materials-15-07576-t002:** The changes of crystal structure for H incorporation into different vacancies of Nb_4_AlC_3_.

Site		Length a (Å)	Length b (Å)	Length c (Å)	Volume (Å^3^)
Initial	Final				
Initial structure		3.15	3.15	24.34	208.62
Nb1-v	2-site	-	-	-	-
Nb2-v	-	-	-	-	-
Al-v	Alv	3.15	3.15	24.09	206.98
C1-v	C1v	3.14	3.14	24.33	208.02
C2-v	C2v	3.14	3.14	24.33	207.78

**Table 3 materials-15-07576-t003:** Cohesive energies for Nb_4_AlC_3_ with different vacancies.

	Length a (Å)	Length b (Å)	Length c (Å)	Volume (Å^3^)	*E_coh_* (eV/atom)
Initial	3.15	3.15	24.34	208.62	−9.24
Nb1-v	3.14	3.14	24.30	207.95	−9.14
Nb2-v	3.14	3.14	24.28	207.56	−9.13
Al-v	3.15	3.15	24.16	207.56	−9.27
C1-v	3.14	3.14	24.30	207.97	−9.20
C2-v	3.14	3.14	24.34	207.78	−9.18

**Table 4 materials-15-07576-t004:** Bond populations and bond lengths for H- Nb(1, 2), Al, or C(1, 2) bonds.

Site	Bond Population					Bond Length (Å)				
	H-Nb1	H-Nb2	H-Al	H-C1	H-C2	H-Nb1	H-Nb2	H-Al	H-C1	H-C2
T-1	0 × 3		0.37		−0.02 × 3	1.976 × 3		1.728		2.651 × 3
T-2	0.20, 0.19		0.23 × 3			2.174, 2.281		1.971 × 3		
O-3	0.22, 0.20 × 2		0.07 × 3		−0.05	2.090, 2.110 × 2		2.230 × 2, 2.260		2.330
T-4	0.11	−0.13 × 3		−0.04	−0.01 × 3	1.830	1.940 × 3		2.02	2.010 × 3
T-6	−0.01 × 3	−0.31 × 3		−0.02 × 3	0.69	2.962 × 3	1.913 × 3		2.360 × 3	1.222
C1 v		0.12 × 6					2.273 × 6			
C2 v	0.14 × 3	0.05 × 3				2.103 × 3	2.334 × 3			
Al v	0.17 × 6					2.813 × 6				

## Data Availability

Not applicable.

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
