# Peer review of "Exploring Hydrogen Incorporation into the Nb4AlC3 MAX Phases: Ab Initio Calculations"

_materials, 2022, doi:10.3390/ma15217576_

Round 1

Reviewer 1 Report

The manuscript by Fu et al performed ab initio calculations to understand the stability and diffusion of the hydrogen atoms at different sites of MAX-phase Nb3AlC3. The calculation method and convergence tests are convincing. The chosen PBE functional is usually blamed for the underestimation of band gaps; however, it is suitable for this study. The results and discussion are well presented. The reviewer believes the findings are important for the study of Nb3AlC3 as a novel hydrogen storage material and are of interest to the readers of Materials. I consider the topic original and relevant in the field. It addresses the theoretical understanding of whether  MAX-phase Nb3AlC3 is a possible candidate for hydrogen storage. Some studies have used ab initio calculations to explore the mechanism of H incorporation into MAX phase materials. However, the study is novel for Nb3AlC3. The methodology is appropriate, and the convergence tests have been carefully considered. However, the presentation should be improved: please refer to my previous comments 1 and 2. The conclusions are consistent with the evidence and arguments presented and they address the main question posed.

If the following questions are properly addressed, the reviewer recommends the manuscript be published in Materials.

1.       The references in the Method section should be updated to use the original work. For example, the reference for MP sampling should be revised to “Monkhorst H J and Pack J D 1976 Phys. Rev. B 13 5188–92”. The GGA work was first described in J. P. Perdew, K. Burke, and M. Ernzerhof, Phys. Rev. Lett. 77, 3865 (1996). In addition, the CASTEP needs to be properly cited.

2.       On line 142, the E_{for} probably is a typo and should be corrected to E^{H-v}_{tot}. In addition, “Where” should be lowercase.

3.       The authors should replace Figure 7 with high-resolution plots. In the current form, it is difficult to read the scale, different DOS projection labels, and sub-plot labels.

4.       In Table 4, there are several format errors. The unit of bond length (a typo band needs also to be corrected) is missing. The format of entries also needs to be corrected. 

Author Response

Point 1: The references in the Method section should be updated to use the original work. For example, the reference for MP sampling should be revised to “Monkhorst H J and Pack J D 1976 Phys. Rev. B 13 5188–92”. The GGA work was first described in J. P. Perdew, K. Burke, and M. Ernzerhof, Phys. Rev. Lett. 77, 3865 (1996). In addition, the CASTEP needs to be properly cited.

Response 1:

Thank you for your comments. We have added the suggested inserted literature and thank the reviewer for this professional comment.

Point 2: On line 142, the E_{for} probably is a typo and should be corrected to E^{H-v_{tot}. In addition, “Where” should be lowercase.

Response 2:

Thank you to the reviewer for this question.  We have corrected this in the manuscript.

Point 3:   The authors should replace Figure 7 with high-resolution plots. In the current form, it is difficult to read the scale, different DOS projection labels, and sub-plot labels.

Response 3:

We thank the reviewer for this suggestion. We have prepared the new Fig.7.

Point 4: In Table 4, there are several format errors. The unit of bond length (a typo band needs also to be corrected) is missing. The format of entries also needs to be corrected.

Response 4:

Your comments are of great help to us in improving the quality of the paper.  We have adjusted Band to Bond, and additionally added the unit of bond length.

Reviewer 2 Report

1)      Table 4. Is it bond length or band length ? Please clarify. The unit of bond population and bond length is missing.

2)      Please use subscript for the required words. For example, Pc (line 349), Pi (line 350), Ti3AlC2 (line 230) and Nb4AlC3 throughout the manuscript.

3)      The units should be mentioned correctly. In line 95, it should be ‘GPa’ and not ‘Gpa’. Line 94, ‘eV’ instead of ‘ev’.

4)      Line 130, what is lager ?

5)      Mention the full form of words like DOS and PDOS when it first appears in the manuscript.

6)      The text in Figure 7 are not readable. Please increase the font size.

7)      Please mention DOI in all the references.

Author Response

Point 1:   Table 4. Is it bond length or band length ? Please clarify. The unit of bond population and bond length is missing.

Response 1:

Your comments are of great help to us in improving the quality of the paper.  We have adjusted Band to Bond, and additionally added the unit of bond length.

Point 2: Please use subscript for the required words. For example, Pc (line 349), P(line 350), Ti3AlC2 (line 230) and Nb4AlC3 throughout the manuscript.

Response 2:

Thank you to the reviewer for this question.  We have corrected this in the manuscript.

Point 3:   The units should be mentioned correctly. In line 95, it should be ‘GPa’ and not ‘Gpa’. Line 94, ‘eV’ instead of ‘ev’.

Response 3:

We thank the reviewer for this suggestion. We have corrected this in the manuscript.

Point 4:   Line 130, what is lager ?

Response 4:

Your comments are of great help to us in improving the quality of the paper.  We have adjusted lager to larger.

Point 5:  Mention the full form of words like DOS and PDOS when it first appears in the manuscript.

Response 5:

Your comments are of great help to us in improving the quality of the paper.  We have corrected this in the manuscript.

Point 6:   The text in Figure 7 are not readable. Please increase the font size.

Response 6:

Your comments are of great help to us in improving the quality of the paper.  We have prepared the new Fig.7.

Point 7:   Please mention DOI in all the references.

Response 7:

Your comments are of great help to us in improving the quality of the paper.  We have corrected this in the manuscript.
